# Effect of 5-Azacitidine Treatment on Redox Status and Inflammatory Condition in MDS Patients

**DOI:** 10.3390/antiox11010139

**Published:** 2022-01-09

**Authors:** Paola Montes, Ana Guerra-Librero, Paloma García, María Elena Cornejo-Calvo, María del Señor López, Tomás de Haro, Laura Martínez-Ruiz, Germaine Escames, Darío Acuña-Castroviejo

**Affiliations:** 1Centro de Investigación Biomédica, Departamento de Fisiología, Facultad de Medicina, Instituto de Biotecnología, Parque Tecnológico de Ciencias de la Salud, Universidad de Granada, 18016 Granada, Spain; paola.montes.sspa@juntadeandalucia.es (P.M.); aguerit@ugr.es (A.G.-L.); lauramartinezr@ugr.es (L.M.-R.); gescames@ugr.es (G.E.); 2UGC de Laboratorios Clínicos, Hospital Universitario Clínico San Cecilio, 18016 Granada, Spain; msenor.lopez.sspa@juntadeandalucia.es (M.d.S.L.); tomas.haro.sspa@juntadeandalucia.es (T.d.H.); 3CIBERfes, Ibs.Granada, 18016 Granada, Spain; 4UGC de Hematología y Hemoterapia, Hospital Universitario Clínico San Cecilio, 18016 Granada, Spain; paloma.garcia.martin.sspa@juntadeandalucia.es (P.G.); mariae.cornejo.sspa@juntadeandalucia.es (M.E.C.-C.)

**Keywords:** myelodysplastic syndrome (MDS), oxidative stress, cytokines, inflammation, 5-azacitidine, somatic alterations

## Abstract

This study focused on the impact of the treatment with the hypomethylating agent 5-azacitidine on the redox status and inflammation in 24 MDS patients. Clinical and genetic features of MDS patients were recorded, and peripheral blood samples were used to determine the activity of the endogenous antioxidant defense system (superoxide dismutase, SOD; catalase, CAT; glutathion peroxidase, GPx; and reductase, GRd, activities), markers of oxidative damage (lipid peroxidation, LPO, and advanced oxidation protein products, AOPP). Moreover, pro-inflammatory cytokines and plasma nitrite plus nitrate levels as markers of inflammation, as well as CoQ10 plasma levels, were also measured. Globally, MDS patients showed less redox status in terms of a reduction in the GSSG/GSH ratio and in the LPO levels, as well as increased CAT activity compared with healthy subjects, with no changes in SOD, GPx, and GRd activities, or AOPP levels. When analyzing the evolution from early to advanced stages of the disease, we found that the GPx activity, GSSG/GSH ratio, LPO, and AOPP increased, with a reduction in CAT. GPx changes were related to the presence of risk factors such as high-risk IPSS-R or mutational score. Moreover, there was an increase in IL-2, IL-6, IL-8, and TNF-α plasma levels, with a further increase of IL-2 and IL-10 from early to advanced stages of the disease. However, we did not observe any association between inflammation and oxidative stress. Finally, 5-azacitidine treatment generated oxidative stress in MDS patients, without affecting inflammation levels, suggesting that oxidative status and inflammation are two independent processes.

## 1. Introduction

Myelodysplastic syndromes (MDS) are a heterogeneous group of clonal hematological disorders clinically characterized by the presence of peripheral blood cytopenias, dysplasia in myeloid lineages, and ineffective hematopoiesis, with a high rate of conversion to acute myeloid leukemia (AML). The annual incidence of the disease is 4–5 cases per 100,000, with an increase of the incidence in individuals age 70 years (30 per 100,000) [1,2].

Currently, factors that determining MDS etiology and its progression to AML are not clarified; however, several authors suggest tumor genetic alterations in genes that regulate cellular proliferation, survival, and differentiation on hematopoietic stem cell [3]. In this line, complex karyotypes and somatic mutations associated with molecular risk are important risk factors related to advances stages of the disease and progression to AML [4,5]. On the other hand, there is considerable evidence that immunity has also be implicated in the pathogenesis of MDS. Recent investigations have identified immunological factors that impair immune surveillance and contribute to clonal immune escape. Related to this, changes affecting the immunogenicity of the CD34+ cell, and a dysregulation of the cellular immune response in the tumor microenvironment, have been observed [6,7].

On the basis of this heterogeneity of factors, clinical course and survival rate of MDS patients is highly variable. Several valuable markers define risk groups and candidates for targeted treatment approaches. The current score system, the Revised International Prognostic Scoring System (R-IPSS), divides patients into several risk prognostic categories [8]. In this line, therapies in low-risk patients consist in blood product transfusions and growth factor treatments while higher-risk patients are treated with hypomethylating agents such us 5-azacitidine (5-AZA) [9].

The evidence that oxidative stress participates in the development and progression of a wide variety of hematological neoplasms is increasing. Leukemic cells are known to live under oxidative stress conditions and they acquire multiple mechanisms to protect themselves from stress, including the induction of antioxidant and detoxifying enzymes [10]. It has been observed that reactive oxygen species (ROS) suppress self-renewal hematopoietic stem cells, which was related to myelodysplasia and ineffective hematopoiesis [11,12]. With regard to MDS, these findings suggested that the excess of ROS could participate not only in the DNA, protein, and lipid oxidative damage, but also in the pathogenesis and drug resistance. In addition, ROS excess induces cytokine imbalance in favor of pro-inflammatory cytokines through NF-κB activation pathway [13], which may support genomic instability and dysplasia of bone marrow precursors, as has been observed in a murine model of MDS [14]. Although several authors have reported the presence of oxidative stress with an upregulation of ROS levels, they have not evaluated the global endogenous antioxidant defense system, and they only have focused on isolated measurement of some biomarkers related to the cellular redox metabolism [15,16,17]. For this reason, in the present study, we carried out a complete evaluation of the entire endogenous antioxidant system by the evaluation of antioxidant defense molecules (disulfide glutathione, GSSG, and glutathione, GSH), the enzymes involved in this defense (superoxide dismutase, SOD; catalase, CAT; glutathione reductase, GRd; and glutathione peroxidase, GPx), and markers of oxidative damage (lipid peroxidation, LPO, and advanced oxidation protein products, AOPP). Additionally, we also evaluated the inflammatory profile expression through the determination of pro-inflammatory cytokines, metabolites of nitric oxide (nitrite plus nitrate), and coenzyme Q10 (CoQ10) plasma levels. In addition, we evaluated for the first time the possible impact of the hypomethylating agent 5-AZA treatment on oxidative stress in MDS patients. Simultaneously, high-risk molecular factors (karyotyes and/or genetic mutations) were analyzed in order to find a possible interplay between them.

## 2. Materials and Methods

### 2.1. Patients and Controls

The study included 24 myelodysplastic syndrome (MDS) patients (12 males, 12 females; mean age of 70 years) of the Granada’s University Hospital (Hospital Universitario Clínico San Cecilio, Granada, Spain) between May 2020 and May 2021. MDS patients were diagnosed on the basis of the World Health Organization (WHO-2016) classification [18] and were registered on the Spanish MDS Registry (RESMD).

According to WHO classification, MDS patients were classified as: early-stage disease (ES) (<5% bone marrow blasts): 2 with single lineage dysplasia (MDS-SLD), 7 with multilineage dysplasia (MDS-MLD), 1 with SLD and ring sideroblasts (MDS-RS-SLD), 2 with MLD and ring sideroblasts (MDS-RS-MLD), 2 with ring sideroblasts (MDS-RS), and 2 with isolated del(5q) MDS (del5q), and advanced-stage disease (AS) (>5% bone marrow blasts): 4 with excess blasts-1 (MDS EB-1) and 4 with excess blasts-2 (MDS EB-2). In terms of prognostic scores, MDS patients were classified according to several risk categories. IPSS-R: low risk (very low risk (VLR)/low risk (LR)) and high risk (intermediate risk (INT)/high risk (HR)/very high risk (VHR)). Cytogenetic risk: favorable (very good (-Y, del(11q), good (normal, del(20q), del(5q) alone or with 1 other anomaly and del(12p)), poor (poor (complex with 3 abnormalities, der(3q) or chromosome 7 abnormalities), very poor (complex with ≥3 abnormalities)), or intermediate (all other single or double abnormalities not listed). Patient demographic and clinical features are summarized in Table 1 and Appendix A.

MDS patients were classified according to the established therapy as untreated MDS patients (patients at diagnosis or only with supportive care (erythropoietin, Epo and/or granulocyte macrophage colony-stimulating factor, GM-CSF)) or treated MDS patients (patients treated with hypomethylating agents such as 5-AZA (from 3 to 7 cycles)). Only one MDS patient received chemotherapy treatment and was included within treated patients. Blood samples of 3 MDS patients were analyzed at the time of diagnosis and after receiving 4 and 7 cycles of 5-AZA (Patient 7 and Patient 19, respectively) and chemotherapy treatment (Patient 21) (Appendix A).

Moreover, we analyzed samples from 20 healthy donors (control group). All controls were volunteer peripheral blood donors with normal hemograms (median age 70 years; range 42–88 years) who had no history of neoplastic disease, previous exposure to chemotherapy drugs, radiation therapy, or immunotherapy.

### 2.2. Blood Samples

Peripheral blood samples were collected by peripheral venipuncture from the antecubital vein in MDS patients and controls between 8 and 10 a.m. Samples were centrifuged at 3000× *g* for 15 min. Plasma was separated and erythrocytes were washed twice with cold saline. Plasma and erythrocytes were aliquoted and stored at −80 °C until the assays were performed.

### 2.3. Measurement of GSH and GSSG Levels

Glutathione (GSH) and glutathione disulfide (GSSG) levels were measured in erythrocytes by using an established fluorometric method and a microplate fluorescence reader (FLx800; Bio-Tek Instruments, Inc., Winoosky, VT, USA) [19]. Results are expressed as µmol/g Hb.

### 2.4. Measurement of GPx, GRd, CAT, and SOD Activities

Glutathione reductasa (GRd), glutathione peroxidase (GPx), catalase (CAT), and superoxide dismutase (SOD) activities were measured in the erythrocyte fraction.

GPx and GRd activities were spectrophotometrically measured, following the NADPH oxidation for 3 min at 340 nm [20] in a 96 well plate spectrophotometer (PowerWaveX; Bio-Tek Instruments, Inc., Winoosky, VT, USA). GRd activity was measured using the kit (703202; Cayman Chemical, Ann Arbor, MI, USA), following the manufacturer’s instructions. CAT activity was measured following the decomposition of H_2_O_2_ at 240 nm according to the Aebi’s method [21]. The enzyme activities were expressed as µmol/min/g Hb. SOD activity was assayed in terms of its ability to inhibit the auto-oxidation of adrenalin to adrenochrome at pH 10.2. Oxidation of adrenalin was measured at 490 nm for 10 min at 30 °C, as described [22]. SOD activity was expressed as U/mg Hb (1 unit = 50% inhibition of auto-oxidation of epinephrine). Hemoglobin concentration was spectrophotometrically determined by the Drabkin’s method [23].

### 2.5. Measurement of LPO, AOPP, and NOx Levels

Products of lipid peroxidation (LPO), advanced oxidation protein products (AOPP), and nitrite plus nitrate levels were measured in plasma samples.

LPO was determined by a commercial kit (KB03002, Bioquochem, Oviedo, Spain), following the manufacturer’s instructions. AOPP were spectrophotometrically measured. Samples were calibrated with chloramine-T solution that in the presence of potassium iodide absorb at 340 nm [24]. A total of 200 μL of plasma diluted 1/5 in PBS, 10 μL of PBS, and 20 μL of concentrated acetic acid were added to sample wells. The standard curve was made with 10 μL of 1.16 M potassium iodide, 200 μL of chloramine-T solution (0–100 nmol/mL), and 20 μL acetic acid. The absorbance of the reaction mixture was immediately read at 340 nm on a microplate reader against a blank. AOPP and LPO concentration was expressed in nmol/mL.

For nitrite + nitrate measurement (NOx), plasma samples were deproteinized with ice-cold 6% sulfosalicylic acid, incubated at room temperature for 30 min and centrifuged at 10,000× *g* for 15 min; then, 50 μL supernatant was incubated with 4 μL 1.25% NaOH, 36 μL of 14 mM phosphate dehydrogenase, 750 μM glucose-6-phosphate, 30 mU nitrate reductase, and 10 μL of a 3 μM NADPH solution for 60 min at room temperature. The concentration of NOx was measured following the Griess reaction, which coverts nitrite into a compound spectrophotometrically detected at 550 nm [25]. Plasma levels of NOx are expressed in mol/L.

### 2.6. Assessment of Cytokine Levels

Cytokines were measured in plasma fraction. The Bio-Plex Pro-Human Cytokine assay kit (BIORAD) was used to analyze the profile expression of pro-inflammatory (IL-1β, IL-2, IL-4, IL-6, IL8, TNF-α, INF-ɣ) and anti-inflammatory (IL-10) cytokines, following the manufacturer’s instructions. Cytokines were analyzed in a Luminex 200 system (Luminex xMAP technology, Luminex Corporation, Austin, TX, USA), and the concentration of each cytokine was determined using xPonent 3.1 Sofware analysis. A parallel standard curve was constructed for each cytokine. Levels are expressed in pg/mL.

### 2.7. Determination of CoQ10 Concentration in Plasma

Plasma CoQ10 extraction was carried out using a hexane/ethanol mixture. CoQ10 levels were determined via reversed-phase UHPLC (UltiMate 3000, Thermo Scientific, Madrid, Spain) coupled to electrochemical (EC) detection, using a mobile phase consisting of methanol, ethanol, 2-propanol, acetic acid (500:500:15:15), and 50 mM sodium acetate at a flow rate of 0.9 mL/min. The electrochemical detector consisted of an ECD-3000RD with the following setting: guard cell (upstream of the injector) at +900 mV, conditioning cell at –600 mV (downstream of the column), followed by the analytical cell at +350 mV. The results were expressed in ng/mL [26].

### 2.8. Statistical Analysis

Data are expressed as the mean ± SEM. Kruskal–Wallis followed by nonparametric-test (Mann–Whitney *U* test) were used to compare the ranks between different groups. Spearman analysis was used to evaluate correlations between quantitative variables. The statistical software GraphPad Prism version 6.0 for Windows (GraphPad Software Inc., La Jolla, CA, USA) was used for all analyses. A two-sided *p*-value < 0.05 was considered statistically significant.

## 3. Results

### 3.1. An Improvement of Oxidative Status WasObserved in Untreated MDS Patients

The characterization of intracellular oxidative level from untreated MDS patients (*n* = 19) was carried out in the erythrocyte fraction by the evaluation of the antioxidant glutathion and antioxidant enzymes. Firstly, a significant lower of GSSG levels (*p* < 0.001) and GSSG·GSH^−1^ ratio (*p* < 0.01) were observed in untreated MDS patients compared to healthy controls (Figure 1A,C). The reduced GSSG·GSH^−1^ ratio was mostly attributable to changes in the GSSG fraction, while intracellular GSH levels were slightly greater (Figure 1B). However, total intracellular glutathione content (GSH + GSSG) did not differ in MDS patients with respect to the control group.

In relation to the antioxidant enzymes, we observed a markedly increase CAT activity compared to controls (*p* < 0.01) (Figure 1D). The rest of the enzymatic activities, GPx (Figure 1E), GRd, and SOD (data not shown), were similar in MDS patients and in controls

Moreover, we evaluated the extracellular oxidative status by the analysis of LPO and AOPP plasma levels. Compared to the control group, MDS patients had a significant decreased of LPO levels (*p* < 0.001) (Figure 1F); however, AOPP levels did not differ between both groups (Figure 1G).

### 3.2. Azacitidine Increased Oxidative Stress in MDS Patients

Firstly, we studied whether 5-AZA treatment might affect the oxidative status in MDS patients. Compared to the untreated MDS group, we observed that the 5-AZA group (*n* = 8)increased the erythrocyte GSSG·GSH^−1^ ratio that was mainly dependent of the levels of GSSG, reaching similar values to the control group (Figure 1A,C), and a reduction in the GSH levels (Figure 1B) in these group of patients (*p* < 0.05). However, we did not observe significant differences in the enzyme activities of the glutathione redox cycle GPx (Figure 1E) and GRd (data not shown) between the 5-AZA and MDS groups. On the other hand, there was a markedly reduced activity of CAT in the 5-AZA group compared to untreated MDS patients (Figure 1D) (*p* < 0.05), showing similar values to the controls. Regarding oxidative markers, LPO values increased twice in the 5-AZA group compared to untreated MDS patients (*p* < 0.05), reaching similar LPO values to controls (Figure 1F). AOPP levels, however, were similar in both untreated and 5 AZA treated groups (Figure 1G).

### 3.3. An Inflammatory Cytokine Microenvironment Was Observed in MDS Patients

It is known that MDS etiology is somewhat related to the interaction between hematopoietic stem cell and their microenvironment. Inflammation is thought to promote tumor development and progression through cytokines, chemokines, and growth factors. In this line, we evaluated the inflammatory microenvironment in 22 peripheral blood samples from MDS patients (untreated MDS (*n* = 15); 5-AZA group (*n* = 7)) through the analysis of pro-inflammatory cytokines, metabolites of NO, and CoQ10 plasma levels. The levels of IL-2, IL-6, and TNF-α were significantly increased in untreated MSD compared to the control group (*p* < 0.001) (Figure 2A,B,E). No significant differences, however, were observed in IL-1β, IL-8, and INF-ɣ levels between untreated MDS group and controls. The anti-inflammatory IL-10 levels increased in untreated MDS patients compared to the control group (*p* < 0.001) (Figure 2D).

Because 5-AZA treatment increases oxidative stress in MSD patients, we asked whether 5-AZA could also induce a pro-inflammatory response. Our results, however, show that although IL-8 levels increased in the 5-AZA group compared to untreated MSD patients (Figure 2C), most of the cytokine levels studied were similar in untreated MDS patients and 5-AZA-treated ones, discarding this hypothesis (Figure 2A,B,D,E). Finally, IL-4 and GM-CSF levels, which were also measured in plasma, were undetectable in MSD patients.

Regarding NOx, we found a significant decrease in plasma of untreated MDS patients and 5-AZA group compared to controls (*p* < 0.05). However, we did not find significant differences between untreated and 5-AZA patients. On the other hand, plasma CoQ10 concentration decreased in untreated MDS group compared to controls (*p* < 0.05), but no differences were detected in CoQ10 levels in untreated MDS and 5 AZA groups.

### 3.4. Redox Balance in MDS Patients during Disease Progression

In order to explore if the changes observed in the redox balance and oxidative status from untreated MDS patients are conserved during the disease progression, we categorized the MDS patient group according to the recent WHO classification, as early stage (ES) (*n* = 13) or advanced stage (AS) (*n* = 6) of the disease. We observed that changes observed in balance redox and antioxidant system defense described above in the MDS group took place from the ES of the disease. In the ES MDS phase, we observed a marked decrease in GSSG levels (*p* < 0.01) with an inversion of the GSSG·GSH^−1^ ratio (*p* < 0.001) compared to controls (Figure 3A), a significant increase in CAT activity (*p* < 0.05) (Figure 3C), and a decrease in LPO and AOPP levels (*p* < 0.001, *p* < 0.05; respectively) (Figure 3D,E). However, as the disease progressed, the GSSG·GSH^−1^ erythrocyte ratio did not differ between both stages of the disease but tended to increase in the advanced stage. Total glutathione levels did not differ between the different phases of the disease.

On the other hand, although we did not previously observe any difference in GPx activity in untreated MDS patients with respect to the control group, we highlight the notable decrease in GPx activity in the early stages of the disease compared to controls (*p* < 0.01); however, this activity increased markedly in the AS with respect to the early stage (*p* < 0.05) (Figure 3B), obtaining values similar to the control group. By contrast, we did not observe differences in the rest of the oxidative enzymes (CAT, SOD, GRd) between both stages of MSD progression. Finally, LPO and AOPP tended to increase in AS of the disease, although they were not significant (Figure 3D,E).

### 3.5. Cytokine Profile Expression in MDS Patients Was Maintained during Disease Progression

Cytokine profile was also analyzed in ES (*n* = 9) and AS (*n* = 6) groups of MSD patients. We observed that IL-2, IL-6, and IL-8 plasma concentration were increased in the ES of the disease with respect to the control group (*p* < 0.001, *p* < 0.001, *p* < 0.05; respectively) (Figure 4A–C); however, while plasma levels of IL-6 and IL-8 cytokines were maintained during the progression of the disease, IL-2 values increased in the AS compared to the early stage (*p* < 0.05) (Figure 4A). No significant differences were observed in the pro-inflammatory cytokines IL-1β and INF-ɣ between both stages of the disease. Regarding IL-10 and TNF-α levels, we observed an increase in the ES compared to controls (*p* < 0.001) (Figure 4D,E). Furthermore, immunosuppressive IL-10 was higher in the AS compared to the early stage (*p* < 0.05) (Figure 4D).

We found decreased in plasma levels of NOx in the ES of untreated MDS compared to controls (*p* < 0.05) (Figure 3F). However, we did not find significant differences between both stages of the disease. In relation to CoQ10, we did not observe differences in its plasma levels between the early and advanced stages of MSD (Figure 3G). Finally, in order to evaluate if it could exit a possible interplay between inflammation and oxidative stress in MDS patients, we analyzed all these factors in untreated MDS group, and we did not observe any association between them.

### 3.6. Oxidative Stress and Risk Prognostic Factors in MDS Patients

We also studied if there was any relationship between the oxidative stress parameters studied and some clinical biological characteristics of MDS patients, including IPSS-R score, karyotype, and mutational profile. For this evaluation, untreated MDS patients were divided into two groups according different categories: high (*n* = 9) vs. low (*n* = 10) IPSS-R risk; cytogenetic risk (favorable (good karyotypes) (*n* = 13) vs. unfavorable (intermediate, poor, and very poor karyotypes) (*n* = 6); molecular risk: high risk (presence of at least one mutation in any of high molecular risk (HMR) genes) (*n* = 11) vs. low risk (absence of mutations in HMR genes) (*n* = 8), and mutational score (number of total mutations present in the dysplastic clone (≤2 (*n* = 13) vs. >3 (*n* = 5) total mutations)).

On the basis of the IPSS-R prognostic stratification, we observed that MDS patients with a high risk of progression to AML presented higher values of GPx activity (*p* < 0.05) (Figure 5A) with a significant decrease in CAT activity (*p* < 0.05) compared to low-risk MDS categories. Similarly, we observed the increase in GPx activity when MDS patients were divided in relation to the score mutational, observing an increase in GPx activity in patients with a high number of total mutations (>3 total mutations) (*p* < 0.01) (Figure 5B) and with an increase in LPO values (*p* < 0.05) with respect to those with low mutational score (≤2 total mutations). In addition, we observed a positive correlation between LPO and number of total mutations (r = 0.594) (*p* < 0.01). Next, when we categorized MDS patients in relation to molecular risk, we also found an increase in GPx activity in the group of patients that presented high risk compared to those who did not present HMR mutations (*p* < 0.05) (Figure 5C). However, when we analyzed oxidative parameters on the basis of cytogenetic risk, we did not find significant differences between both established groups.

### 3.7. Leukocyte Populations in Peripheral Blood of MDS Patients

A significantly lower absolute leukocyte count was observed in untreated MDS patients than in healthy controls (*p* < 0.001) (Figure 5D). In relation to leukocyte populations, neutrophil, monocyte, and lymphocyte counts were also decreased (*p* < 0.001) (Figure 5E,F). On the other hand, relating to 5-AZA group, we observed a significantly lower level of absolute leukocyte count in peripheral blood samples compared to untreated MDS patients (*p* < 0.05) (Figure 5D). The myeloid lineage was altered, with a significantly lower level of absolute neutrophils and monocyte counts (*p* < 0.05) (Figure 5E,F); however, lymphoid lineage cells (absolute lymphocyte count) did not differ significantly between untreated MDS and 5-AZA group.

## 4. Discussion

Myelodysplastic syndromes can be considered as a representative premalignant hematopoietic disorder that can progress towards AML. A number of publications showed the heterogenicity of factors that could be involved in the inadequate hematopoiesis, dysplastic hematopoietic cells, and bone marrow stromal defects [3,4,5,27]. Among these, several studies have found evidence of the involvement of free radicals in a wide variety of hematological neoplasms [28,29]. Oxidative stress occurs when there is an imbalance between reactive oxygen species (ROS) production and the response of the endogenous antioxidant defense systems, resulting in ROS accumulation and cell damage. Regarding MDS, the oxidative condition observed by other authors [11,12,17] led us to investigate the existence of a dysregulation of the endogenous antioxidant defense system that could be related to the development and progression of the disease.

This is the first study in which a complete analysis of the oxidative status was carried out in MDS patients in peripheral blood samples. To date, data published in the literature show an increase in ROS levels with alterations in some markers related to the cellular redox response, mainly GSH levels. However, there is no consensus in the participation of ROS: while some authors described MDS cases with elevated intracellular GSH in bone marrow [11] or neutrophils [15], others reported its decrease in blood plasma [16]. Furthermore, these authors carried out isolated measurements of some oxidative stress parameters, and they did not perform a complete study of the endogenous antioxidant defense system. Thus, they cannot know how the increase of ROS levels affects the redox state. Differences in the expression of antioxidant enzymes among lineages and maturity stages were previously described [30]; therefore, differences observed in the present investigation with other groups in relation to the parameters studied may be due to the type of sample analyzed.

First of all, our results reveal an improvement of intracellular oxidative status in erythrocytes of MDS patients, observing a significant decrease in the GSSG·GSH^−1^ ratio with respect to the control group [31], mainly due to decreasing GSSG levels, while total glutathione levels are conserved in both groups. The decrease in the GSSG·GSH^−1^ ratio and GSSG levels, observed mainly in the early phase of the disease, could be explained by the decrease in GPx activity. By contrast, the control of the hydrogen peroxide detoxification in the early MDS phase was attributed to the increase in the CAT activity and could explain the decrease in the cellular oxidative damage observed, reflected in the decrease in LPO and AOPP plasma levels.

However, as the disease progressed, we observed a tendency of the oxidative stress to increase, reflected by the slight increase in the GSSG·GSH^−1^ ratio in advanced MDS stages. This non-significant increase could be explained by the low number of MDS patients in this group of the disease. Interestingly, as CAT activity decreased, we observed a marked increase in GPx activity, obtaining activity values even above the control group, playing an important role for the elimination of peroxides in the advance stage and reflecting a compensatory mechanism. In this line, GPx response has also been observed for other authors in bone marrow cells from MDS patients with 5% to 19% blast count, where in the antioxidant response was insufficient in eliminating the excess of ROS observed [17]. However, GRd activity did not changed in advances MDS stages, preventing the full regeneration of the GSSG produced by the increased GPx activity. Thus, the reduction of the GRd activity in erythrocytes could explain the tendency to the increase of GSSG·GSH^−1^ ratio and GSSG levels in this group of patients. In this line, we observed an increase of oxidative stress, with a significant increase in LPO and a tendency towards an increase of AOPP levels [32]. The reduced activity of GRd here reported may thus depend on the high production of ROS, which directly damage the enzyme [33]. In turn, the decrease in CoQ10 plasma concentration in MDS patients could be related to the mitochondrial dysfunction and systemic inflammation observed by other groups [34].

On the other hand, different cancer models suggest that high ROS levels contribute to cancer development and progression through genetic mechanisms. Regarding MDS, the oxidative DNA damage product, 7,8-dihydro-8-oxoguanine, has been observed in peripheral leukocytes from MDS patients [35]. For this reason, we analyzed whether the presence of clinical variables and several risk factors, such as mutations in high molecular risk (HRM) genes (TP53, ETV6, ASXL1, RUNX1, EZH2), and/or an increase in the number of“drivers”mutations (≥3 total mutations) could be related with the dysregulation here observed in the antioxidant defense system. Our results revealed, as occurs in advanced stages of the disease, an increase in GPx activity in the group of patients with mutations associated with molecular risk or patients with a greater mutational score. On the basis of these results, the increase in GPx activity could constitute an unfavorable prognostic marker associated with the risk of disease progression. In relation to cytogenetic risk, however, we did not find any association between oxidative stress parameters and karyotypic abnormalities, although and there is an article published in the bibliography that shows a possible association in MDS patients [36].

In relation to deregulation of the inflammatory signaling in MDS, we observed an alteration in the profile expression in MDS patients. As with many other hematological neoplasms, MDS are intimately associated with a dysregulated immune system. Several studies show an altered expression of at least 30 cytokines both in the medullary microenvironment and peripheral blood samples [37]. In general, our results reveal an upregulation in the concentration of plasma TNF-α, IL-2, IL-6, and IL-8 levels in MDS patients, reflecting an alteration of the inflammatory signaling. These results agree with other reports that show an increase of these cytokine values in bone marrow and peripheral blood from MDS patients [37,38]. In this line, high levels of TNF-α have been related to apoptosis rates in MDS bone marrow cells, and they have been also considered adverse prognostic factor [39,40]. On the other hand, when we analyze by MDS subtypes, low- and high- counts of blasts according to WHO classification, we observed a significant increase in IL-2 concentration in advanced stages, while TNF-α, IL-6, and IL-8 plasma levels did not differ between both stages of the disease. Other studies, however, showed a downregulation of these cytokines in high-risk cases [41]. Moreover, the immunosuppressive cytokine IL-10 has also been observed to be elevated in MDS patients, and its concentration was higher in advanced MDS stages, a finding also reported by other authors in high-risk MDS patients [41].

Nevertheless, there are some remarkable discrepancies between the results of this work and previous studies in relation to INF-ɣ and IL-1β values. In the present report, we did not find an increase in these cytokines in MDS patients compared to controls. In relation to INF-ɣ, these authors observed an overexpression in marrow mononuclear cells, but in our study, in plasma did not yield any change [42]. Our data could be explained due to the marked lymphopenia T previous observed in MDS patients, together with the greater presence of exhausted CD8 T cells in peripheral blood samples [7]. Finally, we did not find any association between oxidative stress parameters and pro- and anti-inflammatory cytokines studied. Thus, according to our results, these markers appear to make an independent contribution to the development and progression of the disease.

Finally, therapy in MDS patients is based on the risk, and, in highest-risk disease, the goal is to prolong survival. Therapy in this group of MDS patients is based on hypomethylating agents (HMA), such as 5-azacitidine (5-AZA). In this line, we evaluated whether treatment with 5-AZAcould affect oxidative stress parameters and if it could be related with an increase of inflammation. It has been observed that 5-AZA shows a range of biological effects, including mutagenic, leukopenic, immunosuppressive, and antineoplastic activity [43]. Regarding oxidative stress, decitabine, the other hypometilant agent used in the treatment of MDS, has been observed to induce ROS accumulation [44]. However, in relation to MDS, there is only one publication in the literature and it displays controversial results [45]. For this reason, in the present investigation, we evaluated for the first time the possible role of oxidative stress in MDS patients with and without 5-AZA treatment. Our results revealed an increase of GSSG·GSH^−1^ ratio with respect to untreated MDS patients, with an increase in GSSG levels and a significant decrease in GSH concentration. In this context, GRd does not increase its activity, and thus reduced glutathione is not regenerated. Likewise, the decrease in the CAT activity in the 5-AZA group is related to a marked increase in LPO plasma concentration. Therefore, on the basis of our results, we found that 5-AZA treatment increases oxidative stress in blood samples from MDS patients. In this line, use of antioxidants with 5-AZA could improve the response of the patients to the treatment. Because 5-AZA has been reported that 5-AZA upregulates the MT1 melatonin receptors in some tumors [46], and given the antioxidant properties of melatonin, further studies with melatonin plus 5-AZA therapies would be of interest.

## 5. Conclusions

Alteration of redox status and the presence of a pro-inflammatory environment are two independent factors involved in the pathogenesis of MDS. On the other hand, 5-AZA treatment initiates oxidative stress in MDS patients.

## 6. Limitations of the Study

We are conscient that the number of patients is low, mainly due to the requirements for the inclusion in the study and the relative low percentage of MDS patients. Thus, although we found some trends of the markers measured in our study, it would be possible to detect significant changes by increasing the number of patients.

## Figures and Tables

**Figure 1 antioxidants-11-00139-f001:**
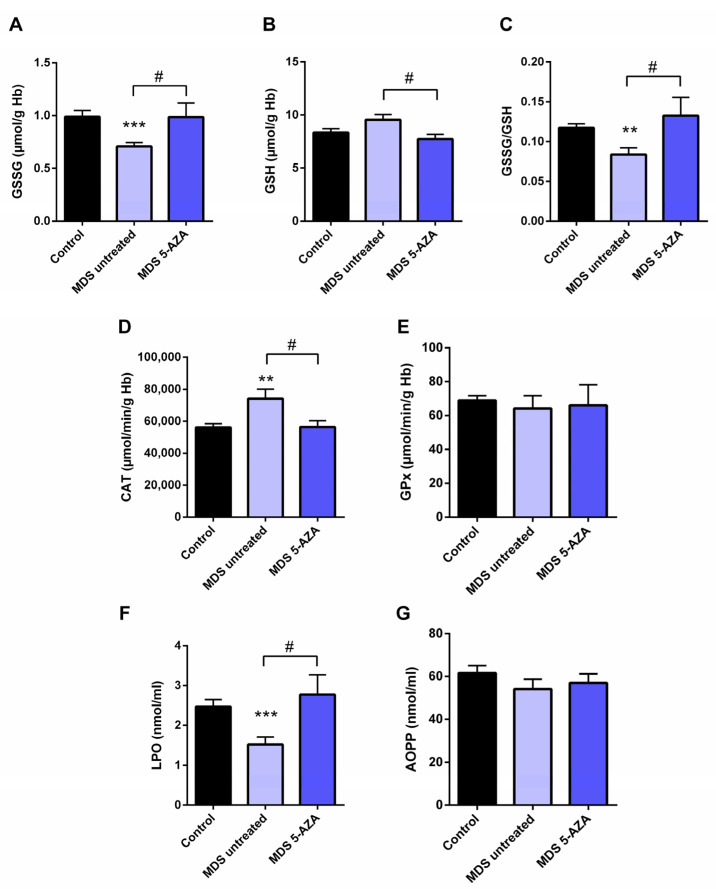
Analysis of erythrocyte and plasmatic oxidative stress parameters in myelodysplastic syndrome (MDS) patients and controls. MDS patients were divided at diagnosis or only with supportive care (untreated MDS) or MDS patients treated with 5-azacitidine (5-AZA MDS). The following oxidative stress levels are represented: (**A**) erythrocyte levels of oxidized glutathione (GSSG); (**B**) reduced glutathione (GSH); (**C**) GSSG·GSH^−1^ ratio; (**D**) catalase (CAT) and (**E**) glutathione peroxidase (GPx) activity; (**F**) lipid peroxidation (LPO) and (**G**) advanced oxidation protein products (AOPP) plasma levels. Data are presented as mean ± SEM. ** *p* < 0.01, *** *p* < 0.001 vs. control; # *p* < 0.05 vs. 5-AZA MDS group.

**Figure 2 antioxidants-11-00139-f002:**
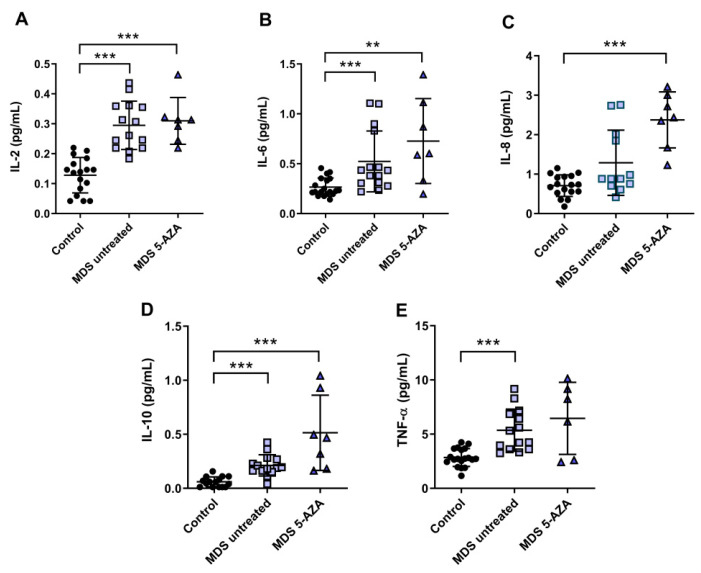
Concentration of pro- and anti-inflammatory cytokines in plasma samples from myelodysplastic syndrome (MDS) patients. (**A**) IL-2, (**B**) IL-6, (**C**) IL-8, (**D**) IL-10 and (**E**) TNF-α plasma concentration of MDS patients at diagnosis or with supportive care (untreated MDS) and MDS patients treated with 5-azacitidine (5-AZA MDS). Data are presented as mean ± SEM. ** *p* < 0.01, *** *p* < 0.001 vs. control.

**Figure 3 antioxidants-11-00139-f003:**
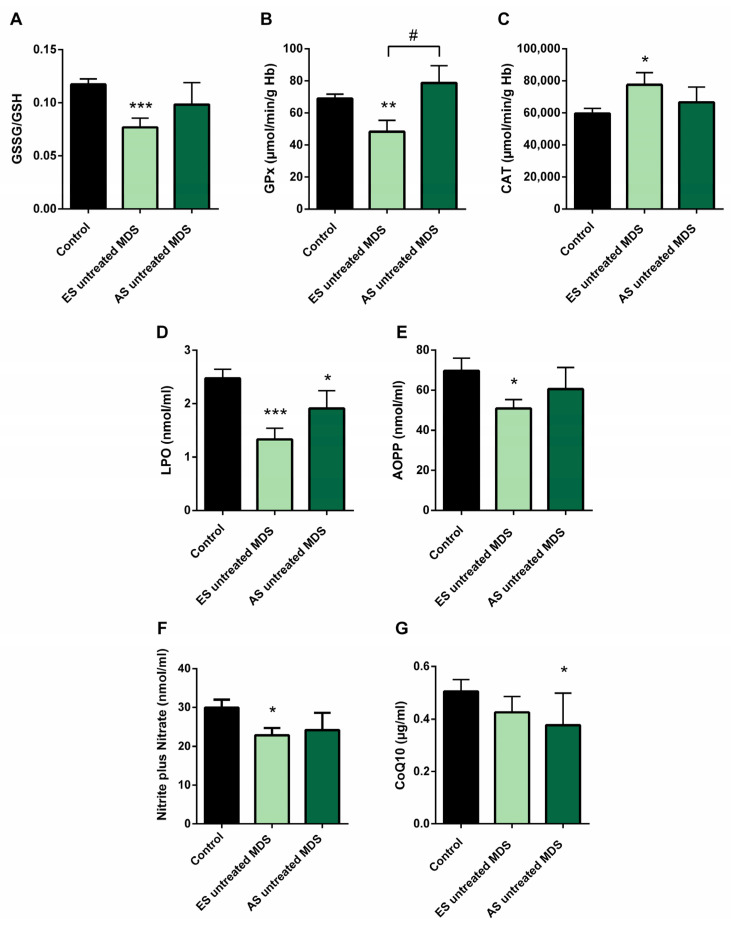
Oxidative stress biomarkers during myelodysplastic syndrome (MDS) progression. (**A**) erythrocyte GSSG·GSH^−1^ ratio; (**B**) glutathione peroxidase (GPx) and (**C**) catalase (CAT) activities; (**D**) lipid peroxidation (LPO); (**E**) advanced oxidation protein products (AOPP); (**F**) nitrite plus nitrate; and (**G**) CoQ10 levels in plasma in early stage (ES untreated MDS) vs. advanced stage of untreated MDS patients (AS untreated MDS). Data are presented as mean ± SEM. * *p* < 0.05, ** *p* < 0.01, *** *p* < 0.001 vs. control; # *p* < 0.05 vs. ES untreated MDS group.

**Figure 4 antioxidants-11-00139-f004:**
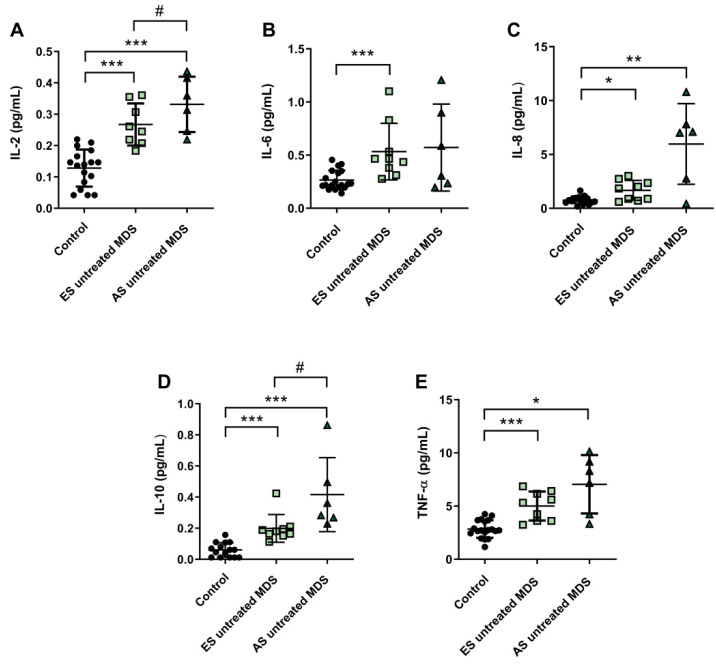
Concentration of pro- and anti-inflammatory cytokines during myelodysplastic syndrome (MDS) progression. (**A**) IL-2, (**B**) IL-6, (**C**) IL-8, (**D**) IL-10 and (**E**) TNF-α plasma concentration in early stage (ES untreated MDS) vs. advanced stage (AS untreated MDS) from untreated MDS patients. * *p* < 0.05, ** *p* < 0.01, *** *p* < 0.001 vs. control; # *p* < 0.05 vs. ES untreated MDS group.

**Figure 5 antioxidants-11-00139-f005:**
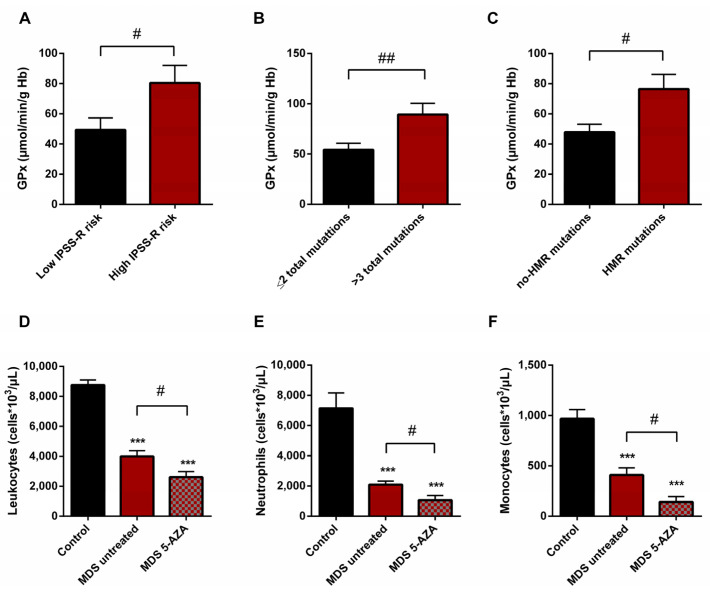
Risk prognostic factors and leucocyte populations in MDS patients. GPx activity in untreated MDS patients according to (**A**) IPSS-R score, (**B**) mutational score and (**C**) molecular risk. (**D**) Leucocyte populations, (**E**) neutrophils and (**F**) monocytes in untreated MDS patients and 5-AZA group. # *p* < 0.05, ## *p* < 0.01; *** *p* < 0.001 vs. control.

**Table 1 antioxidants-11-00139-t001:** Biodemographic and clinical characteristics of myelysplastic syndrome (MDS) patients. MDS-SLD: MDS with single-lineage dysplasia; MDS-MLD: MDS with multilineage dysplasia; MDS with SLD and ring sideroblasts (MDS-RS-SLD); MDS with MLD and ring sideroblasts (MDS-RS-MLD); MDS-RS: MDS and ring sideroblasts; MDS del(5q): MDS with isolated del(5q); MDS EB-1, -2: MDS with excess of blasts-1;-2.IPSS-R: Revised International Prognostic Scoring System; VHR: very high risk; HR: high risk; INT: intermediate; LR: low risk; VL: very low risk. Cytogenetic risk: favorable (very good (−Y, del(11q), good (normal, del(20q), del(5q) alone or with 1 other anomaly and del(12p)), poor (poor (complex with 3 abnormalities, der(3q) or chromosome 7 abnormalities), very poor (complex with ≥3 abnormalities)), or intermediate (all other single or double abnormalities not listed).

MDS Patients	
No.	24
Mean age (years)	70 (42–88)
Sex	12 male/12 female
WHO-2016 classification	MDS-SLD (8.3%)
	MDS-RS-SLD (4.2%)
	MDS-MLD (29.2%)
	MDS-RS-MLD (8.3%)
	MDS-RS (4.2%)
	MDS del(5q) (8.3%)
	MDS EB-1 (20,8%)
	MDS EB-2 (16.7%)
IPSS-R risk groups	Low risk (50%) (very low (VL)/low (LR))
	High risk (50%) (intermediate (INT)/high (HR)/very high (VHR))
Cytogenetic	Favorable (75%)
	Intermediate (8.3%)
	Poor (16.7%)

## Data Availability

The data presented in this study are available on request from the corresponding author due to data privacy and confidentiality of the patients.

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
