# Peer review of "Effect of 5-Azacitidine Treatment on Redox Status and Inflammatory Condition in MDS Patients"

_antioxidants, 2022, doi:10.3390/antiox11010139_

Round 1
Reviewer 1 Report
Paola Montes et al. in manuscript entitled “Effect of 5-Azacitine treatment on redox status and inflammatory condition in MDS patients” presented a complete analysis of the oxidative status in patients with myelodysplastic syndrome (MDS). Authors also evaluated the possible impact of the hypomethylating agent 5-AZA treatment on oxidative stress in MDS patients. It is ery important and interesting work; nevertheless some conclusions are not clear for me and it is my major comment. How authors understand their statement in discussion “First of all, our results reveal an improvement of intracellular oxidative status in erythrocytes of MDS patients, observing a significant decrease in the GSSG·GSH−1 ratio with respect to the control group [31], mainly due to decreasing GSSG levels, while total glutathione levels are conserved in both groups. “? Also the statement “Nevertheless, there are some remarkable discrepancies between the results of this work and previous studies in relation to INF-É£ and IL-1βvalues. In the present report, we did not find an increase in these cytokines in MDS patients compared to controls. In relation to INF-É£, these authors observed an overexpression in marrow mononuclear cells but or study, in plasma did not yield any change [42].” is unclear as in the MS I cannot find (in M&M and results) anything about INF-É£ and IL-1β analyses?
I have also a few minor comments:
- All the abbreviations used in text should be presented at the beginning in form of the list to make it easier for readers.
- There are a lot of different qualifications of the patients. It could be clearer to present it in the form of table or diagram with overlapping different treatment (or add this information to the supplement Table 1).
- The Fig. 1 description diagram G should be in capital letter.
- In figure 2 it is difficult to know between which groups the results are statistically significant.
- Page 7, line 261, please change “CoO10 concentration decreased” to “CoQ10 concentration decreased”
- 3 and results in point 3.4 should reflect redox balance in MDS patients during disease progression but without diagram of patients’ qualification (please see comment 2nd) it is difficult to know the number of analysed patients.
- The figure (or panels in Fig.3) with NOx and CoQ10 should be added to MS.
- Also graphic representations for the points 3.6 and 3.7 should be added to MS.
Author Response
Paola Montes et al. in manuscript entitled “Effect of 5-Azacitine treatment on redox status and inflammatory condition in MDS patients” presented a complete analysis of the oxidative status in patients with myelodysplastic syndrome (MDS). Authors also evaluated the possible impact of the hypomethylating agent 5-AZA treatment on oxidative stress in MDS patients.
Thanks very much for your comments. Please find below pour responses.
Major points:
1) How authors understand their statement in discussion “First of all, our results reveal an improvement of intracellular oxidative status in erythrocytes of MDS patients, observing a significant decrease in the GSSG·GSH−1 ratio with respect to the control group [31], mainly due to decreasing GSSG levels, while total glutathione levels are conserved in both groups. “?
Oxidized/reduced glutathione ratio is the main marker of intracellular oxidative stress, as has been previously reflected by other authors: "Jones, D.P. Redox potential of GSH/GSSG couple: assay and biological significance. Methods Enzymol.2002, 348, 93–112, doi:10.1016/s0076-6879(02)48630-2". The glutathione cycle is a combination of the activities of glutathione peroxidase and reductase, and changes in the activity of these enzymes may modify the balance between the two forms of GSH without changes in the total glutathione. In our results, the absence of changes in GPx activity in MDS group prevents the consumption of GSH to form GSSG and, thus, the ratio GSSG·GSH−1 could be unchanged. However, during ES of the disease, GPX activity decreased, suggesting a reduction in the consumption of GSH, thus reducing the levels of GSSG. This changes may prevail in untreated MDS patients, reflecting the decrease in the GSSG·GSH−1 ratio.
2) Also the statement “Nevertheless, there are some remarkable discrepancies between the results of this work and previous studies in relation to INF-É£ and IL-1β values. In the present report, we did not find an increase in these cytokines in MDS patients compared to controls. In relation to INF-É£, these authors observed an overexpression in marrow mononuclear cells but or study, in plasma did not yield any change [42].” Is unclear as in the MS I cannot find (in M&M and results) anything about INF-É£ and IL-1β analyses? "
In the section “2.6. Assessment of cytokine levels" of the Material and Method section, cytokine analysis also includes the determination of IL-1B and INF-É£ (page 5, lines 192-199).
In the section “3.3. An inflammatory cytokine microenvironment is observed in MDS patients”, we commented that "No significant differences, however, were observed in IL-1β, IL-8 and INF-É£ levels between untreated MDS group and controls", (page 7, new revised manuscript).
In the section “3.5. Cytokines profile expression in MDS patients is maintained during disease progression”, we commented that "No significant differences were observed in the pro-inflammatory cytokines IL-1β and INF-É£ between both stages of the disease", (page 11).
Minor points:
1) All the abbreviations used in text should be presented at the beginning in form of the list to make it easier for readers.
We appreciate this suggestion. All abbreviations used in text have been included at the beginning of the revised manuscript (page 1, abbreviations section, new revised manuscript).
2) There are a lot of different qualifications of the patients. It could be clearer to present it in the form of table or diagram with overlapping different treatment (or add this information to the supplement Table 1).
We agree with the reviewer. We have added this information to the supplement Table 1. We have incorporated the following parameters into the table: IPSS-R risk, citogenetic risk, score mutational and molecular risk of the 24 patients included in the study. Also, table S1 description diagram have been modified.
In addition, details of the number of patients analyzed in each section have been incorporated in the text (pages 5, 6, 7, 9, 11, 13; revised manuscript).
3) The Fig. 1 description diagram G should be in capital letter.
Corrected.
4) In figure 2 it is difficult to know between which groups the results are statistically significant.
To facilitate comprehension, Figure 2 and also, Figure 4, have been modified accordingly. We have added top lines that identify the different groups (new revised manuscript).
5-Page 7, line 261, please change “CoO10 concentration decreased” to “CoQ10 concentration decreased”.
Corrected.
6) 3 and results in point 3.4 should reflect redox balance in MDS patients during disease progression but without diagram of patients’ qualification (please see comment 2nd) it is difficult to know the number of analysed patients.”
As mentioned above, we have incorporated in the text the number of patients analyzed in each section.
7) The figure (or panels in Fig.3) with NOx and CoQ10 should be added to MS.
We appreciate this suggestion. We have included panels (F) and (G) in Figure 3, corresponding to NOx and CoQ10 plasma levels, and we have incorporated the descriptions (F) and (G) in the Fig. 3 description diagram.
Furthermore, and we have also added "Figure 3G" and "Figure 3F" in the section 3.5. "Cytokines profile expression in MDS patients is maintained during disease progression" in the text of the revised manuscript (page 10, lines 337-339).
8) Also graphic representations for the points 3.6 and 3.7 should be added to MS.
We have added a new figure in the manuscript (Figure 5, page 11), which includes the results obtained from points 3.6 and 3.7; and the Figure 5 description diagram. Also, we refer to Figure 5 A-F in sections 3.6 and 3.7 of the text (pages 10-11).
Reviewer 2 Report
In the manuscript presented here the authors evaluated oxidative stress markers as well as inflammatory cytokine expression in regards to 5-azacytidine treatment of MDS patients. The work is carefully and well done, measuring the most important oxidative stress factors and the basic pro- and anti- inflammatory cytokines. Interestingly the inflammatory cytokines are maintained during disease progression and the authors show here that 5-azacitidine increases oxidative stress in MDS patients.
Minor comments:
Title: 5 - Azacitine correct into 5-Azacitidine
Figure 1.These data should be individualised
Fig. 2. Is the statistically higher level of Il-10 MDS-5AZA due to only 2 samples? Please comment on it.
Author Response
In the manuscript presented here the authors evaluated oxidative stress markers as well as inflammatory cytokine expression in regards to 5-azacytidine treatment of MDS patients. The work is carefully and well done, measuring the most important oxidative stress factors and the basic pro- and anti- inflammatory cytokines. Interestingly the inflammatory cytokines are maintained during disease progression and the authors show here that 5-azacitidine increases oxidative stress in MDS patients.
Thanks very much for your comments. Please find below our responses to your suggestios
Minor points:
1) Title: 5 - Azacitine correct into 5-Azacitidine.
We agree with the reviewer. We have replaced the previous title: ” Effect of 5-Azacitine treatment on redox status and inflammatory condition in MDS patients” with “Effect of 5-Azacitidine treatment on redox status and inflammatory condition in MDS patients”.
2) Figure 1. These data should be individualised.
Due to a better visualization of the results, and less dispersion of the data, the Figure 1 probably can be maintained in bar graphs. For this reason, figures 2 and 4 are individualized because larger dispersion of the data.
3) Fig. 2. Is the statistically higher level of Il-10 MDS-5AZA due to only 2 samples? Please comment on it.
Anti-inflammatory IL-10 levels are statistically higher in MDS-5AZA group compared to controls, but there are not observed differences respect to untreated MDS patients.